Feature selection for emotion recognition in speech: a comparative study of filter and wrapper methods

Altheneyan Alaa
Alhadlaq Aseel asalhadlaq@ksu.edu.sa
Department of Computer Science and Engineering, College of Applied Studies, King Saud University , Riyadh , Saudi Arabia
Saibene Aurora
Electronic publication date: 2025 Sep 16
Publication date: 2025
Volume: 11
Electronic Location ID: e3180
Received 2025 Feb 13; Accepted 2025 Aug 11
Copyright: © 2025 Altheneyan and Alhadlaq
Copyright year: 2025
Copyright holder: Altheneyan and Alhadlaq
License: This is an open access article distributed under the terms of the Creative Commons Attribution License, which permits unrestricted use, distribution, reproduction and adaptation in any medium and for any purpose provided that it is properly attributed. For attribution, the original author(s), title, publication source (PeerJ Computer Science) and either DOI or URL of the article must be cited.
License URL: https://creativecommons.org/licenses/by/4.0/

Keywords: Speech emotion recognition, Machine learning, Artificial intelligence, Pattern recognition, Feature extraction

Funding: King Saud University, Riyadh, Saudi Arabia ORF-2025-796 This research was funded by Ongoing research funding program, (ORF-2025-796), King Saud University, Riyadh, Saudi Arabia. The funders had no role in study design, data collection and analysis, decision to publish, or preparation of the manuscript.

==============================
Feature selection is essential for enhancing the performance and reducing the complexity of speech emotion recognition models. This article evaluates various feature selection methods, including correlation-based (CB), mutual information (MI), and recursive feature elimination (RFE), against baseline approaches using three different feature sets: (1) all available features (Mel-frequency cepstral coefficients (MFCC), root mean square energy (RMS), zero crossing rate (ZCR), chromagram, spectral centroid frequency (SCF), Tonnetz, Mel spectrogram, and spectral bandwidth), totaling 170 features; (2) a five-feature subset (MFCC, RMS, ZCR, Chromagram, and Mel spectrogram), totaling 163 features; and (3) a six-feature subset (MFCC, RMS, ZCR, SCF, Tonnetz, and Mel spectrogram), totaling 157 features. Methods are compared based on precision, recall, F1-score, accuracy, and the number of features selected. Results show that using all features yields an accuracy of 61.42%, but often includes irrelevant data. MI with 120 features achieves the highest performance, with precision, recall, F1-score, and accuracy at 65%, 65%, 65%, and 64.71%, respectively. CB methods with moderate thresholds also perform well, balancing simplicity and accuracy. RFE methods improve consistently with more features, stabilizing around 120 features.

Introduction

Emotion recognition from speech has become a critical aspect of human-computer interaction, affective computing, and social robotics (Ottoni, Ottoni & Cerqueira, 2023). It has become an important criterion to understand the emotional content conveyed through spoken language, as it enables systems to respond appropriately to users’ affective states, resulting in more effective communication and interaction experiences (Toyoshima et al., 2023; Kim & Lee, 2023). Speech emotion recognition (SER) has been applied across various sectors, including customer services, entertainment, home automation, and even in the medical domain (Zhao & Shu, 2023).

One of the essential components of a successful SER system is the use of appropriate features. Traditional emotion recognition methods leverage extracting acoustic features from speech signals, which are then classified using machine learning techniques (Wani et al., 2021). Several studies have investigated the use of Mel-frequency cepstral coefficients (MFCC) (Ottoni, Ottoni & Cerqueira, 2023; Rezapour Mashhadi & Osei-Bonsu, 2023; Likitha et al., 2017; Kumbhar & Bhandari, 2019), root mean square (RMS) value (Ottoni, Ottoni & Cerqueira, 2023; Rezapour Mashhadi & Osei-Bonsu, 2023; Patni et al., 2021), zero crossing rate (ZCR) (Ottoni, Ottoni & Cerqueira, 2023; Rezapour Mashhadi & Osei-Bonsu, 2023), chromagram (Ottoni, Ottoni & Cerqueira, 2023), spectral centroid frequency (SCF) (Rezapour Mashhadi & Osei-Bonsu, 2023), Tonnetz (Rezapour Mashhadi & Osei-Bonsu, 2023), and Mel spectrogram (Ottoni, Ottoni & Cerqueira, 2023; Rezapour Mashhadi & Osei-Bonsu, 2023; Pentari, Kafentzis & Tsiknakis, 2024) as features for emotion recognition. However, not all features extracted from speech are equally informative for distinguishing between different emotional states (George & Muhamed Ilyas, 2024). As a result, feature selection techniques are crucial for identifying the most relevant and discriminative features, which can improve the performance and efficiency of emotion recognition models (Langari, Marvi & Zahedi, 2020).

Recently, various feature selection methods have been proposed to address the challenge of selecting optimal features for various detection tasks. Two main feature selection methods have been explored extensively in literature: filter methods and wrapper-based approaches (Yadav et al., 2023; Milton & Selvi, 2015). Filter methods, such as correlation-based feature selection and mutual information, evaluate the relevance of features independently of the classification model (Milton & Selvi, 2015; Wu et al., 2013; Kovács, Tóth & Van Compernolle, 2015). In contrast, wrapper-based approaches like the recursive feature elimination (RFE) iteratively select features based on their importance in a predictive model (Abdusalomov et al., 2022; El Aboudi & Benhlima, 2016).

Problem statement

Despite the growing interest in emotion recognition, there remains a need for comprehensive comparative studies to evaluate the effectiveness of different feature selection methods. This is because most feature selection approaches have been extensively tested on other speech recognition tasks (Kovács, Tóth & Van Compernolle, 2015). There is a need to define the best features and feature selection methods for SER tasks. Therefore, this article aims to bridge this gap by comparing the performance of filter methods and RFE for speech feature selection in emotion recognition. Through empirical experiments and statistical analysis, we aim to determine the strengths and limitations of each feature selection approach, providing valuable insights for researchers and practitioners in the field of affective computing and human-computer interaction.

Research objectives

The following are the main objectives of this study: To benchmark the performance of filter-based and wrapper-based feature selection methods in SER.

To assess the impact of feature selection techniques on classifier performance using evaluation metrics such as accuracy, precision, recall, and F1-score.

To evaluate the effect of different groups of features (as used in previous studies) on model performance and feature selection outcomes.

To provide insights into selecting appropriate feature selection techniques for enhanced emotion recognition from speech.

To contribute to advancing research in affective computing and human-computer interaction by identifying optimal feature selection methods for emotion recognition applications.

Contribution of the study

This study made the following contributions: Conducting empirical experiments on three different datasets including the Toronto emotional speech set (TESS) (Pichora-Fuller & Dupuis, 2020), crowd-sourced emotional multimodal actors dataset (CREMA-D) (Cao et al., 2014), and Ryerson audio-visual database of emotional speech and song (RAVDESS) (Livingstone & Russo, 2018), to evaluate the performance of the filter and wrapper-based feature selection methods in emotion recognition from speech.

Considering the numerous features employed in previous literature, such as MFCC, RMS, ZCR, chromagram, SCF, Tonnetz, and Mel spectrogram, to ensure a comprehensive evaluation of feature selection methods.

Employing filter approaches like correlation-based feature selection and mutual information, and wrapper approaches like the RFE, to conduct extensive experiments and provide comparisons.

Providing valuable insights for researchers and practitioners in affective computing and human-computer interaction through statistical analysis and interpretation of experimental results, showcasing the effectiveness of various feature selection methods in emotion recognition from speech datasets.

This article is organized as follows: ‘Related Works’ discusses the relevant research on speech emotion recognition, the respective features employed, and feature selection approaches. ‘Methodology’ presents the features, feature selection methods, datasets, and models employed for the experiments. In ‘Results and Discussion’, the experimental results are presented and discussed, while the final thoughts and conclusions are presented in ‘Conclusions’.

Related works

The task of SER holds significant importance across various industries and applications. Consequently, extensive research efforts are continually directed toward advancing novel recognition techniques (Ottoni, Ottoni & Cerqueira, 2023). A critical factor influencing the success of SER systems is the choice of speech features, prompting researchers to extensively investigate and compare different feature types. For instance, Toyoshima et al. (2023) utilized Mel spectrograms in conjunction with features derived from the Geneva Minimalistic Acoustic Parameter Set (GeMAPS) to perform SER experiments. In another study, Kim & Lee (2023) improved the accuracy of emotion recognition by applying dimensionality reduction techniques to visualize emotion-specific audio characteristics. They introduced a hybrid deep learning model that combines a bidirectional long short-term memory (BiLSTM)–Transformer with a 2D convolutional neural network (CNN). The BiLSTM–Transformer component captures temporal speech patterns, while the 2D CNN processes the Mel-spectrogram to extract spatial audio features. Their model demonstrated robust performance, achieving unweighted accuracy scores of 95.65% and 80.19% across different datasets, highlighting the effectiveness of integrating transformer-based models with suitable feature selection in SER tasks.

Zhao & Shu (2023) presented an improved emotion analysis in speech using combined spectro-temporal modulation (STM) and entropy features incorporated into CNN utilized to reduce the dimensions of these features and extract the features of each signal. The proposal tested on the Berlin and ShEMO datasets shows an average accuracy of 93.33%, and 85.73%, respectively, outperforming other methods by 6.67%. Similarly, another study by Likitha et al. (2017) conducted SER using MFCC features, focusing on “Happy”, “Sad”, and “Angry” emotions. Their system achieved approximately 80% accuracy, reinforcing the effectiveness of MFCC features in capturing emotional cues in speech. The results provided by these two studies showed impressive performance. Efforts were made in Patni et al. (2021) and Pentari, Kafentzis & Tsiknakis (2024) to improve these performances by employing other features. For instance, Patni et al. (2021) used a two-dimensional (2D) CNN to classify emotions from speech utilizing features such as the MFCC, gammatone frequency cepstrum coefficient (GFCC), energy, pitch, and chromatogram achieving an accuracy that is 9% better than similar works of Mustaqeem & Kwon (2020). However, limiting their works to a few datasets confirms the need for more empirical experiments on three or more datasets, and exploring the filter and wrapper-based feature selection methods is yet to be fully studied.

Langari, Marvi & Zahedi (2020) extracted new features (adaptive time-frequency features) using the fractional Fourier transform which was combined with cepstral features to capture a robust classification of different emotional classes in three datasets—EMO-DB, SAVEE, and PDREC. However, the critical question yet to be answered is ‘Should the motivation for feature extraction be solely to achieve higher accuracy rather than the need to capture the relevant feature that can guarantee speaker-independent speech recognition?’

Milton & Selvi (2015) proposed a four-stage feature selection method using the divide and conquer principle to reduce the computational complexity of SERs. The challenge, however, was sacrificing accuracy while reducing computational complexity. Also, the validation of their proposal using different feature grouping under various scenarios such as variation in the number of stages is yet to be explored.

Kovács, Tóth & Van Compernolle (2015) argued against researchers employing different feature selection methods to find the best filter set when using the Gabor filters. They proposed using heuristic rules to construct filter sets manually. The drawback of their approach is the lack of guarantee of optimality which would have been a strong motivation for providing an alternative to the automatic filter selection methods. Also, manually selecting features defeats the goal of automation and its attendant benefits which SERs promote.

Abdusalomov et al. (2022) on the other hand provided a machine learning-based feature parameter extraction. Their approach maximized and effectively optimized the use of cache memories in SERs. The limitation, however, is their inability to curtail the errors from various noise environments. Researchers have suggested that such challenges can be reduced by reducing the number of features in the dataset (Kacur et al., 2021). Also, improving the accuracy of most SERs is an ongoing research effort. Another area of difference between their work and this article is that while Abdusalomov et al. (2022) based their work on MFCC, this article is premised on MFCC, RMS, ZCR, chromagram, SCF, Tonnetz, and Mel spectrogram yielding a more robust comparison and validated results.

Hossain, Khan & Kader (2024) envisioned a speech recognition model exploiting non-invasive electroencephalogram (EEG) signals and machine learning to assist speech-impaired people. Unlike most works that relied on public datasets, they developed a dataset of preprocessed EEG signals of the most frequently used English alphabets and digits. While their work focused on dataset development and investigating the best machine learning applicable, the work gave insight into the undeniable impact of features, feature extraction, and processing to guarantee successful speech recognition solutions.

More recent works show continued efforts in enhancing SER systems using diverse methods. Kumar et al. (2024) proposed a deep learning-based SER architecture designed for real-world applications. Although their work emphasizes practical deployment, it does not elaborate on specific datasets or feature engineering approaches, limiting reproducibility. Barhoumi & BenAyed (2024) developed a real-time SER system combining MFCC, ZCR, RMS, Chroma, and Mel spectrogram features with MLP, CNN, and CNN-BiLSTM models. They used spectrogram shifting and background noise injection for data augmentation across TESS, EmoDB, and RAVDESS, showing promise for noisy environments.

Du et al. (2025) addressed low-accuracy challenges in SER by introducing a hybrid CNN-SNN architecture. They proposed a perceptual neuron encoding layer (PNEL) to capture both static and dynamic emotion features from IEMOCAP, enhancing emotion recognition under biologically plausible computation. A similar study by Chowdhury, Ramanna & Kotecha (2025) proposed a lightweight ensemble architecture combining CNN and CNN-BiLSTM using hand-crafted features like ZCR, RMSE, Chroma STFT, and MFCC. Another study by Aghajani & Zohrevandi (2025) proposed a novel architecture using three parallel CNN-LSTM branches that process MFCC, Mel spectrogram, and temporal-frequency domain features. Integrated through a self-attention mechanism, their model demonstrated improved generalisation in emotion recognition tasks, though specific datasets were not disclosed.

These recent studies illustrate the evolving domain of SER, highlighting advancements in deep learning, multimodal features, and diverse dataset usage. However, several technical gaps remain. Most works focus on limited acoustic features, often excluding spectral, tonal, or statistical measures that could improve performance. Additionally, the majority do not explore or compare wrapper and filter-based feature selection techniques, leaving their practical utility unclear. Validation is also frequently confined to one or two datasets, limiting generalizability. This study addresses these gaps by employing a broad feature set, evaluating three feature selection methods, CB, MI, and RFE, and validating across three benchmark datasets: TESS, CREMA-D, and RAVDESS. Table 1 provides a comprehensive summary of the related studies.

Table 1 Summary of related works comparing datasets, features, AI models and consideration for wrapper vs. filter approach performance.

Article	Datasets	Features	AI model	Compare wrapper and filter approach?	
Toyoshima et al. (2023)	IEMOCAP	Mel, GeMAPS	CNN	No	
Kim & Lee (2023)	EMO-DB, RAVDESS	Mel	BiLSTM, 2D CNN	No	
Zhao & Shu (2023)	Berlin, ShEMO	STM Entropy	CNN GC ECOC	No	
Likitha et al. (2017)	Private	MFCC	Wavread	No	
Patni et al. (2021)	RAVDESS	MFCC, GFCC, Chroma, RMS	2D-CNN	No	
Langari, Marvi & Zahedi (2020)	EMO-DB, SAVEE, PDREC	ADFT, Cepstral, MFCC	LIB SVM	No	
Milton & Selvi (2015)	Berlin, eNTERFACE	MFCC, Formants, ZCR	Pool of classifiers	No	
Abdusalomov et al. (2022)	Private	MFCC	KNN	No	
Kumar et al. (2024)	Not specified	Not specified	Deep learning	No	
Barhoumi & BenAyed (2024)	TESS, EmoDB, RAVDESS	MFCC, ZCR, RMS, Chroma, Mel Spectrogram	MLP, CNN, CNN-BiLSTM	No	
Du et al. (2025)	IEMOCAP	Not explicitly stated	CNN, SNN	No	
Chowdhury, Ramanna & Kotecha (2025)	RAVDESS, TESS, SAVEE, CREMA-D, EmoDB	ZCR, RMSE, Chroma STFT, MFCC	CNN, CNN-BiLSTM	No	
Aghajani & Zohrevandi (2025)	Not specified	MFCC, Mel-spectrogram, Temp-Freq Features	CNN-LSTM, Self-attention	No	
This article	TESS, CREMA-D, RAVDESS	MFCC, RMS, ZCR, SCF, SB, Tonnetz, Mel, Chroma	CNN	Yes CB, MI, and RFE	

Methodology

The main objective of this study is to apply filter and wrapper approaches for feature selection, utilizing speech features previously investigated in prior research. To ensure the reproducibility of this study, a clear workflow illustrating the steps taken is provided in Fig. 1. This workflow commences with the selection of datasets to be utilized to evaluate the performance of our methods. Specifically, the TESS (https://utoronto.scholaris.ca/collections/036db644-9790-4ed0-90cc-be1dfb8a4b66), CREMA-D (https://github.com/CheyneyComputerScience/CREMA-D) and RAVDESS (https://zenodo.org/record/1188976) databases were chosen. Subsequently, various data augmentation techniques are applied to the dataset, including noise, pitch, and stretch. Next, feature extraction methods such as MFCC, RMS, ZCR, chromagram, SCF, Tonnetz, Mel spectrogram, and spectral bandwidth (SB) are employed. Subsequently, three distinct feature selection approaches are used including confusion matrix (CM) and mutual information (MI), filter-based approaches, and RFE, wrapper-based approaches. Finally, one deep learning model from existing literature, a 4-block CNN model (Patni et al., 2021) was utilized. The following subsections will delve into the specific steps of the proposed approach in greater detail.

Figure 1 Flowchart of the methodology employed in this study.

Data collection and preprocessing

Three datasets were utilized in this study to ensure diversity and a more robust SER: (i) TESS, and (ii) CREMA-D, and (iii) RAVDESS.

TESS

The TESS dataset includes recordings of two English actresses, one aged 26 and the other 64. The audio recordings are two seconds in length featuring anger, disgust, fear, happiness, neutrality, surprise, and sadness. The dataset includes 2,800 audio files, with 400 files allocated to each emotion category (Pichora-Fuller & Dupuis, 2020; University of Toronto, 2025).

CREMA-D

The CREMA-D dataset includes 7,442 unique audio samples from 91 actors of different racial and ethnic backgrounds (Science, 2025). The dataset comprises 48 male actors and 43 female actors, each uttering 12 sentences. The audio recordings are two seconds in length and express six different emotions: anger, happiness, neutrality, disgust, sadness, and fear (Cao et al., 2014).

Table 2 reflects the final distribution of emotion samples used in this study before augmentation. The TESS dataset contributes an even 400 samples per emotion for seven categories, excluding “calm” (Pichora-Fuller & Dupuis, 2020; University of Toronto, 2025). CREMA-D contains varied samples across six emotions, excluding “calm” and “surprise” (Cao et al., 2014), while RAVDESS provides all eight emotions including the only instances of “calm” and “surprise” (Livingstone & Russo, 2018). These distributions collectively form the merged dataset shown in Fig. 2.

Table 2 Distribution of emotion samples across datasets.

Emotion	TESS	CREMA-D	RAVDESS	Total	
Fear	400	857	606	1,863	
Surprise	400	0	192	592	
Happy	400	857	606	1,863	
Sad	400	857	606	1,863	
Neutral	400	857	326	1,583	
Angry	400	857	606	1,863	
Calm	0	0	192	192	
Disgust	400	857	606	1,863	
Total	2,800	5,142	3,840	11,702	

Figure 2 Histogram showing the distribution of audio samples per emotion category across the combined dataset (TESS, CREMA-D, and RAVDESS).

RAVDESS

The RAVDESS dataset comprises 24 professional actors, 12 male and 12 female, vocalizing two lexically matched statements in a neutral North American accent (Livingstone & Russo, 2018). The dataset includes seven emotions: “calm”, “happy”, “sad” “surprise”, “angry”, “disgust”, and “fearful”. Each emotion is presented at two levels of intensity- (normal, strong) and a neutral expression. The dataset includes audio recordings in three modalities: audio-only, audio-video, and video-only. The dataset contains 2,076 audio recordings with an average length of three seconds per recording (Livingstone & Russo, 2018).

Data analysis

To create a unified dataset for analysis, all audio recordings from the three datasets (TESS, CREMA-D, and RAVDESS) were standardized. Specifically, each audio clip was resampled to a common sampling rate of 16 kHz to ensure consistency. Audio lengths were normalized to 2 s by trimming or zero-padding as needed. Metadata was harmonized, and a consolidated dataset was formed for feature extraction and model training. The statistics of the resulting combined dataset are illustrated in Fig. 2.

Figure 3 presents a comparative visual analysis of audio features for “happy” and “sad” emotional states, depicted through waveplots and spectrograms at different scales. The waveplot of an audio file visually represents the audio signal’s amplitude variations over time. This plot is typically used to display the structure of an audio track at a glance, showing where the sound is loud or quiet, the start and end of different sections, and where silence occurs within the file. The waveplot of the audio data is represented in Figs. 3A and 3D for the “happy” and “sad” emotion categories, respectively. These waveplots show notable differences in the amplitude envelope of the audio signals. The waveform associated with happiness demonstrates a greater amplitude variability, which may indicate a more dynamic and expressive speech pattern commonly found in joyful communication. In contrast, the waveform for sadness displays a flatter envelope, suggesting a more monotone and subdued vocal expression.

Figure 3 Comparative visual analysis of speech characteristics for happy and sad emotions.

(A) Waveplot of happy emotion showing higher amplitude and energy. (B) Linear-scale spectrogram of happy emotion highlighting strong high-frequency components. (C) Log-scale spectrogram of happy emotion emphasizing harmonic structures. (D) Waveplot of sad emotion with reduced amplitude variation. (E) Linear-scale spectrogram of sad emotion showing weaker high-frequency activity and stronger low-frequency concentration. (F) Log-scale spectrogram of sad emotion highlighting low-frequency dominance.

Figures 3B and 3E represent the short-time Fourier transform (STFT) of the happy and sad emotions on a linear scale. These figures reveal the frequency content of the happy and sad emotions on a linear scale. The happy emotion spectrogram has a broader spread of energy across the frequency spectrum, particularly noticeable in the mid to high-frequency range, which corresponds to a more energetic articulation. This contrast is stark against the sad emotion spectrogram, where the energy is more confined and less varied, reflecting the typically lower and narrower pitch range observed in sad speech.

The STFT is an analysis technique used to determine the sinusoidal frequency and phase content of local sections of a signal as it changes over time. This is particularly useful for audio signals, where we must analyze how frequencies vary with time. STFT works by dividing the signal into shorter segments of equal length and computing the Fourier transform separately on each segment. This results in a two-dimensional representation of the signal, where one dimension is time and the other is frequency, with the amplitude given by the magnitude of the transform (Kehtarnavaz, 2008). The STFT of a speech sample can be represented on either a linear or logarithmic scale. Each frequency bin is spaced equally across the frequency axis on a linear scale. This type of scaling is straightforward and gives a uniform spectrum view, which is good for general purposes, particularly where higher frequency resolution is needed across the spectrum. In a logarithmic scale, each octave (doubling of frequency) has the same space on the axis, which means higher frequencies are compressed, and lower frequencies are expanded in the visual representation. This scaling is more aligned with how humans perceive sound, where we tend to notice differences in lower frequencies more than in higher ones (Krishnan, 2021).

Figures 3C and 3F represent these emotions on a log scale, and allow for a deeper inspection of the lower frequencies, which are crucial for identifying speech formants. The happy emotion spectrogram on a log scale shows a richer harmonic structure, with clearly defined formants that suggest a greater degree of vocal tract resonance. In comparison, the sad emotion spectrogram displays less pronounced formants, implying a reduction in resonance which can be attributed to a more closed or constricted vocal tract during sad expressions.

Data augmentation

To expand the dataset and increase model robustness, we applied four augmentation techniques, including adding noise, time stretching, shifting, and pitch alteration, to the preprocessed, combined dataset (TESS, CREMA-D, and RAVDESS). Each original audio sample was used to generate two additional augmented versions, resulting in a threefold dataset expansion. A normal audio sample is illustrated in Fig. 4A.

Figure 4 Visual comparison of waveforms under different audio data augmentation techniques.

(A) Normal audio waveform. (B) Audio with added random noise. (C) Stretched audio signal with slower playback rate. (D) Pitch-shifted audio signal. (E) Time-shifted audio signal.

Adding noise

To add noise to the audio data, we defined a function that receives the audio array as input and adds random noise before returning a noisy data array as output. A random factor of between 0 and 0.035 is used. The noise amplitude is directly proportional to the random factor and the maximum value of the data. A noisy audio data is represented in Fig. 4B.

Stretching

The stretch function was designed to either elongate or condense an array of audio data temporally, dependent on a specified rate factor. When the rate factor is set below 1, the audio is elongated, resulting in a slower playback. Conversely, a rate factor above 1 will condense the audio, increasing the playback speed. In our specific application, we have employed a rate factor of 0.7 to slow down the audio data. Figure 4C represents the stretched audio sample.

Pitch

To alter the pitch of the audio data, we defined a function that takes in the audio array, a pitch factor, and the sampling rate as input and alters the frequency or pitch of the data. The pitch factor determines how much the pitch should be altered. When this value is above 1, it implies that the pitch is increased; when it is below 1, it is decreased. The output is an audio data array, pitch-shifted by a pitch factor of 0.7. Figure 4D represents audio sample with pitch-shift.

Shifting

The shift function, on the other hand, is responsible for temporarily displacing an audio data array by a randomly determined interval. This interval is the product of a random integer, selected from the range of −5 to 5, and the factor 1,000. The displacement is achieved through the numpy.roll function, which cyclically shifts the elements of the array. The outcome of this function is the time-shifted audio data array represented in Fig. 4E.

In the data augmentation phase, two augmentation techniques were applied per sample, resulting in three total variants per instance (original + two augmentations). This tripled the dataset size from 11,682 to 35,046 samples. The original combined dataset contained varying numbers of samples per emotion category, as shown in Fig. 2. We retained all eight emotion categories: fear, happy, sad, neutral, angry, disgust, surprise, and calm. Notably, calm and surprise were underrepresented with 192 and 592 samples, respectively. Table 3 presents the number of samples before and after augmentation for each emotion.

Table 3 Sample distribution per emotion class before and after augmentation ( ×3).

Emotion	Original samples	After augmentation ( ×3)	
Fear	1,863	5,589	
Surprise	592	1,776	
Happy	1,863	5,589	
Sad	1,863	5,589	
Neutral	1,583	4,749	
Angry	1,863	5,589	
Calm	192	576	
Disgust	1,863	5,589	
Total	11,682	35,046	

Feature extraction methods

In this study, we have considered eight different features from the literature, including MFCC, ZCR, RMS, Chroma, Mel, SCF, Tonnetz, and SB. These features capture the speech’s various parts regarding temporal and spectral characteristics needed for emotion recognition. The Librosa library in Python was used to extract these features (Fime, Ashikuzzaman & Aziz, 2024). The subsequent discussion will introduce each of these features. 1. MFCC: These are arguably the most prominent features in speech and audio processing. They offer a representation of the short-term power spectrum of sound based on a linear cosine transform of a log power spectrum on a nonlinear mel scale of frequency (Kumbhar & Bhandari, 2019). MFCCs are computed through several steps: (a) A fast Fourier transform (FFT) is applied to a windowed signal segment to obtain the frequency spectrum.

(b) The spectrum is then mapped onto the mel scale, a perceptual scale that mimics the human ear’s response to different frequencies.

(c) The logarithm of the powers at each mel frequency is taken to capture the magnitude.

(d) A discrete cosine transform (DCT) is applied to the log mel powers to obtain the cepstral coefficients.

The Mel scale is defined by the formula (Fime, Ashikuzzaman & Aziz, 2024): (1) m=2595log10(1+f700)

where m is the perceived frequency in mel scale and f is the actual frequency in Hz.

The k-th MFCC (where k ranges from 1 to the number of coefficients) is given by the equation: (2) MFCC[k]=∑n=0N−1log(S[n])⋅cos⁡[k(n−12)πN]

where S[n] represents the log power at each of the mel frequencies, and N is the number of mel frequency bands.

The MFCC feature has proven effective in capturing the vocal tract configuration, which changes with articulated emotions. As a result, it is invaluable for recognizing distinct emotional states in speech (Kim & Lee, 2023).

2. ZCR: This measures the number of times the audio signal crosses the zero axis. It effectively captures the signal’s frequency content in the time domain and is often used to differentiate percussive sounds from harmonic sounds (George & Muhamed Ilyas, 2024). The ZCR is calculated using the equation: (3) ZCR=12N∑n=1N|sgn(x[n])−sgn(x[n−1])|

where sgn is the sign function, and x[n] is the signal value at sample n, with N being the total number of samples.

3. RMS: The RMS energy of an audio signal measures its power, reflecting the signal’s loudness. This feature is especially useful for detecting emotional intensity in speech.

The RMS is given by the equation: (4) RMS=1N∑n=1Nx[n]2

where x[n] is the signal value at sample n, and N is the total number of samples.

4. Chroma: Chroma features contain the distribution of energy across the 12 different pitch classes and are crucial in capturing the harmonic and melodic characteristics of music. These features are also valuable in speech analysis as they can highlight tonal content that may indicate specific emotional states (Patni et al., 2021).

The computation of Chroma features typically involves several steps: (a) A Fourier transform is applied to the audio to obtain the frequency spectrum.

(b) The spectrum is then mapped onto the musical octave, grouping the energies into 12 bins, each representing one of the 12 semitones (or pitch classes) in a chromatic scale.

The mathematical representation of Chroma features can be expressed as: (5) C(p)=∑kM(k)⋅χp(f(k))

where: • C(p) is the energy in the p-th Chroma bin.

• M(k) is the magnitude at the k-th bin of the Fourier transformed signal.

• f(k) is the frequency corresponding to the k-th bin.

• χp(f(k)) is a function that maps the frequency f(k) to the pitch class p. This function returns 1 if f(k) falls into pitch class p, and 0 otherwise.

This projection simplifies the complex spectrum of frequencies into a more manageable form for analyzing the audio’s tonal aspects, making it particularly useful for identifying melodic and harmonic patterns that may correspond with emotional expressions in speech.

5. Mel spectrogram: The Mel-scaled spectrogram is a spectrogram that converts the frequencies to the Mel scale, emphasizing perceptually relevant frequencies. It is related to MFCCs but retains more detailed frequency information.

It can be visualized by the equation for the mel filter bank: (6) Hm(k)={0fork<f[m−1]k−f[m−1]f[m]−f[m−1]forf[m−1]≤k<f[m]1fork=f[m]f[m+1]−kf[m+1]−f[m]forf[m]<k≤f[m+1]0fork>f[m+1]

where Hm(k) is the filter bank amplitude at frequency bin k, and f[m] is the center frequency of the m-th mel filter.

6. SCF: The Spectral Centroid Frequency represents the ‘center of mass’ of the spectrum and is used to characterize the brightness of a sound. It is computed as the weighted mean of the frequencies present in the sound (George & Muhamed Ilyas, 2024; Patni et al., 2021).

The SCF is calculated using: (7) SCF=∑k=1Nf[k]⋅S[k]∑k=1NS[k]

where f[k] is the frequency in Hz corresponding to the k-th bin, S[k] is the spectral magnitude at bin k, and N is the total number of bins.

7. Tonnetz: The Tonnetz, or tonal centroid features, represent a classical concept in music theory, often visualized as a geometric arrangement of notes that emphasizes their harmonic relationships. These features have been adapted in audio signal processing to analyze the harmonic content and its transitions, which can indicate emotional expressions in speech (George & Muhamed Ilyas, 2024).

These features are computed by first extracting the Chroma features, then applying a transformation that captures the harmonic relations. The transformation reflects the geometric properties of the Tonnetz where notes are arranged in a lattice that reflects fifth and minor third intervals. This allows for capturing the connectivity and movement between tones, which is relevant for understanding musical texture and emotional nuances in speech.

The computation can be expressed in a simplified form as follows: (8) T=H⋅C

where: • T represents the Tonnetz features,

• H is a transformation matrix encoding the harmonic relations between notes,

• C is the vector of Chroma features.

This transformation typically focuses on relations such as perfect fifths and minor thirds, which are significant in Western music theory for defining tonality. By analyzing how these intervals change over time, we can infer harmonic tension and resolution changes, which may correspond to emotional states in speech.

The use of Tonnetz features in emotion recognition reflects the importance of understanding not just the presence of specific pitches but how they relate to each other in a dynamic temporal context (George & Muhamed Ilyas, 2024). This relationship mapping helps identify subtle variations in speech that may be indicative of underlying emotional states.

8. SB: The spectral bandwidth measures the width of the band of frequencies at which significant energy is present around the spectral centroid, indicating the sound’s timbral complexity. (9) SB=(∑k=1N(f[k]−SCF)2⋅S[k]∑k=1NS[k])1/2

where f[k], S[k], SCF, and N are as previously defined.

Feature selection algorithms in SER typically retain features that are highly correlated with emotional variations while discarding those that provide redundant or less discriminative information. For example, MFCC and Mel spectrogram-based features are often preserved because they capture detailed spectral and perceptual characteristics that strongly reflect vocal tract changes associated with emotions (Kumbhar & Bhandari, 2019; Kim & Lee, 2023; Fime, Ashikuzzaman & Aziz, 2024). Similarly, energy-based features such as RMS and pitch-related measures like ZCR are frequently retained, as they provide cues about emotional intensity and arousal (George & Muhamed Ilyas, 2024). On the other hand, features like spectral bandwidth or certain Tonnetz components may be removed when they exhibit low mutual information with the target emotion classes or when they overlap in contribution with other dominant features (Kim & Lee, 2023). This selective process helps reduce dimensionality while maintaining the most informative subset for emotion classification.

Feature selection methods

Feature selection plays a crucial role in the development of machine learning models for emotion recognition. By identifying and utilizing a subset of relevant features, this process enhances model interpretability, reduces dimensionality, and potentially improves model performance by eliminating irrelevant or redundant data. The choice of feature selection method often depends on the characteristics of the data and the requirements of the specific model employed.

Filter method

Filter methods are employed to assess the relevance of features based on statistical measures, providing scores for each feature. These scores are used to rank features, selecting those most likely to be useful for model training or removing those that do not meet a predefined relevance threshold. This approach is especially efficient as it is applied as a preprocessing step, independent of the model training phase.

Correlation-based (CB) feature selection: CB feature selection is particularly effective in the context of emotion recognition as it identifies features that show a strong correlation with the target emotional states but minimal correlation with each other. This helps reduce multicollinearity and improve the generalizability of the model.

The effectiveness of a feature set can be evaluated using a heuristic that considers both the individual predictive ability of each feature and the redundancy among them. The merit of a feature subset S containing k features can be computed as:

(10) Ms=krcf¯k+k(k−1)rff¯

where rcf¯ is the average correlation of the features with the class, and rff¯ is the average inter-correlation among the features.

Mutual information (MI): MI is used to quantify how much information one feature shares with the target variable, making it an invaluable tool in determining feature relevance without assuming linear relationships. For emotion recognition, MI can highlight non-linear dependencies that are vital for understanding complex emotional patterns.

The mutual information between a feature X and the target variable Y is calculated as:

(11) I(X;Y)=∑x∈X∑y∈Yp(x,y)log⁡(p(x,y)(y))

where p(x,y) is the joint probability distribution of X and Y, and p(x) and p(y) are the marginal distributions.

Wrapper method

Wrapper methods involve a search for the optimal subset of features that maximizes a specific model’s performance criterion. Unlike filter methods, wrappers evaluate features based on the model’s accuracy, making them more computationally intensive but potentially more accurate.

Recursive feature elimination (RFE): RFE is well-suited for emotion recognition as it iteratively refines the feature set to retain only those that contribute most significantly to model performance. By integrating RFE, emotion recognition models can focus on the most impactful features, potentially enhancing both accuracy and efficiency.

RFE operates through the following steps: Initialize the model on the full set of features and assess their importance.

Eliminate the least important feature(s) based on the model’s criteria.

Re-train the model on the reduced set of features.

Repeat the process until a predetermined number of features is left.

Employing these feature selection strategies helps streamline the modeling process, particularly in complex datasets typical of emotion recognition tasks. By focusing on the most relevant features, these methods can reduce overfitting and improve the predictive power of the models.

Experimental setup

System configuration

The experimental system is configured with a high-performance 64-bit architecture designed for intensive data processing. It features symmetric multiprocessing (SMP) and vsyscall32 capabilities. The system includes 130 GB of system memory, crucial for managing large datasets and complex algorithms efficiently, preventing performance bottlenecks. The central processing unit (CPU) is an Intel® Core™ i9-10940X operating at 3.30 GHz with a capability to reach up to 4,800 MHz. This robust CPU setup supports high-speed processing and multitasking, making it highly suitable for demanding machine learning and data analysis tasks.

Implementation details

The dataset was split using a random shuffling strategy applied to individual audio samples, and by a proportion of 70:15:15 for the train, test, and validation sets, respectively. We designed and trained a CNN model structured using a sequence of convolutional and pooling layers. The first layer is a 1D convolution layer with 256 filters, a kernel size of 5, a stride of 1, and ‘same’ padding. It uses the rectified linear unit (ReLU) as its activation function. A max-pooling layer follows the first convolution layer, with a pool size of 5 and a stride of 2, also with ‘same’ padding. These layers repeat the structure of a Conv1D followed by a MaxPooling1D but with different numbers of filters. The second set also has 256 filters, followed by a set with 128 filters and then 64 filters. This configuration allows the network to capture hierarchies of features at different levels of abstraction. Dropout layers are interspersed after some of the pooling layers, with dropout rates of 0.2 and 0.3. A flatten layer flattens the output from the previous convolutional and pooling layers to form a single long feature vector needed for the dense layers that follow. A fully connected layer with 32 units and ReLU activation function is used to interpret the features extracted by the convolutions and pooling. The final output layer has 8 units with a softmax activation function. The model uses the Adam optimizer, a popular choice for deep learning applications. The loss function is categorical cross-entropy.

Evaluation metrics

Several evaluation metrics were utilized to assess the effectiveness of the emotion recognition solutions developed in this study. These metrics provide insights into different facets of model performance, from overall accuracy to the balance between precision and recall. Below, we detail each of these metrics. Accuracy (ACC) is one of the most intuitive performance measures (Bown, 2024). It is defined as the ratio of correctly predicted observations to the total observations: (12) ACC=TP+TNTP+TN+FP+FN

where TP, TN, FP, and FN represent the true positives, true negatives, false positives, and false negatives, respectively. High accuracy indicates that the model performs well across all classes. However, accuracy alone can be misleading, especially in datasets with imbalanced classes.

Precision (P), or positive predictive value, measures the accuracy of positive predictions (Foody, 2023). Formally, it is defined as the ratio of correctly predicted positive observations to the total predicted positives: (13) P=TPTP+FP

High precision indicates a low rate of false positives, which is crucial in applications where the cost of a false positive is high.

Recall (R), or sensitivity, measures the ability of a model to find all the relevant cases within a dataset (Foody, 2023). It is defined as the ratio of correctly predicted positive observations to all observations in the actual class. (14) R=TPTP+FN

High recall is particularly important in situations where missing a positive instance is significantly detrimental.

The F1-score (F1) is the weighted average of precision and recall. Therefore, this score takes both false positives and false negatives into account (Foody, 2023). It is especially useful when the class distribution is uneven. The F1-score is defined as: (15) F1=2⋅P⋅RP+R

The F1-score is a better measure to use when seeking a balance between precision and recall, and there is an uneven class distribution.

Support is the number of actual occurrences of the class in the specified dataset. While not a performance measure in itself, support provides valuable context when comparing the quality of classification reports across different classes (Foody, 2023). It is crucial for assessing the reliability of the metrics above, particularly in datasets like ours, where some classes (e.g., calm and surprise) are significantly underrepresented.

Results and discussion

This section presents the results of the experiments conducted, including the filter method, wrapper-based results, and a comparative analysis of both approaches. All datasets were concatenated before extracting the various features. The goal of the feature selection task, was to select the most relevant features for the task of emotion recognition.

Baselines

Three main approaches were established as baselines. The first baseline is a CNN model trained on all features generated by the full set of feature extraction methods: MFCC, RMS, ZCR, chromagram, SCF, Tonnetz, Mel spectrogram, and spectral bandwidth, resulting in 170 features. The second baseline is a CNN trained on features generated from five different feature extraction methods, including MFCC, RMS, ZCR, chromagram, and Mel spectrogram, as conducted in Ottoni, Ottoni & Cerqueira (2023). The result of this baseline approach yielded 163 features. The third baseline is a CNN model trained on features generated from six feature extraction methods, including MFCC, RMS, ZCR, SCF, Tonnetz, and Mel, as conducted in Rezapour Mashhadi & Osei-Bonsu (2023). The result of this baseline approach yielded 157 features.

Filter method results

Correlation-based method

The following thresholds were used to evaluate the CB method: 0.02, 0.05, 0.1, and 0.2. These thresholds helped determine the number of features to be selected, based on the CB algorithm. Figures 5A, 5B, and 5C represent the confusion matrix of the emotion recognition result based on the CB method for thresholds 0.05, 0.02, and 0.1, respectively. The best result was obtained when the threshold was set to 0.02, resulting in a selection of the best 158 features and an accuracy of 60.45%.

Figure 5 Confusion matrix for correlation-based feature extraction.

(A) Threshold = 0.05. (B) Threshold = 0.02. (C) Threshold = 0.1.

Table 4 provides more insight into the precision (P), recall (R), and F1-score for the different thresholds. From observation, the performance varies across different thresholds for each emotion category, indicating that the choice of threshold, significantly impacts the performance of the feature extraction method. Different emotion categories exhibit varying performance across the thresholds. For example, the “Surprise” category generally achieves higher precision, recall, and F1-score compared to other categories across all thresholds. In some cases, adjusting the threshold leads to improvements or deteriorations in performance metrics. For instance, for the “Angry” category, lowering the threshold from 0.1 to 0.02 results in an increase in precision and F1-score, albeit a slight decrease in recall. The performance of emotion categories like “Neutral” and “Sad” tends to be lower compared to others, which might indicate class imbalance issues or inherent difficulty in distinguishing these emotions from speech features alone.

Table 4 Performance comparison of Correlation-based feature extraction method at various thresholds.

Threshold	0.1	0.05	0.02		
Metric	P	R	F1	P	R	F1	P	R	F1	S	
Angry	0.72	0.77	0.75	0.76	0.73	0.74	0.73	0.73	0.73	808	
Calm	0.60	0.70	0.64	0.70	0.70	0.70	0.76	0.55	0.64	93	
Disgust	0.52	0.47	0.50	0.50	0.49	0.49	0.53	0.52	0.52	843	
Fear	0.63	0.49	0.56	0.60	0.53	0.56	0.61	0.51	0.56	851	
Happy	0.57	0.54	0.55	0.60	0.58	0.59	0.55	0.55	0.55	865	
Neutral	0.54	0.57	0.55	0.53	0.56	0.54	0.52	0.58	0.55	709	
Sad	0.52	0.66	0.58	0.56	0.63	0.59	0.58	0.65	0.61	809	
Surprise	0.84	0.82	0.83	0.82	0.88	0.85	0.88	0.90	0.89	279	

One more observation is that lower correlation thresholds (e.g., 0.02) tend to result in a broader selection of features, potentially capturing more diverse information but also introducing more noise. Higher thresholds (e.g., 0.1) are more conservative, selecting fewer features but potentially more relevant ones. Also, the sensitivity of performance to changes in the correlation threshold varies across emotions. For example, “Surprise” appears to be less sensitive, maintaining relatively high performance across different thresholds, while “Calm” exhibits more variability.

In this approach, the best performance was obtained at a threshold of 0.02 on the “Surprise” emotion, with a precision score of 0.88, a recall of 0.90, and an F1-score of 0.89. The poorest performance was observed at a threshold of 0.05 on the “Disgust” emotion with a precision score of 0.50, a recall of 0.49, and an F1-score of 0.49.

Mutual information feature extraction method

In implementing the mutual information approach, different values of a ‘K’ factor need to be defined. In this study, K represents the number of features selected from the full feature set based on their mutual information scores with the emotion labels. Five different values were employed: 50, 80, 100, 120, and 150. Figure 6 represents the confusion matrix for these various values of K, with the best performance occurring at K = 120, with a test accuracy of 60.53%, and 120 features.

Figure 6 Confusion matrix for mutual information feature extraction for K = 50, 80, 100, 120, and 150, respectively.

(A) K = 50 features. (B) K = 80 features. (C) K = 100 features. (D) K = 120 features. (E) K = 150 features.

The results of the confusion matrices are elaborated in Table 5, and shows various observations. Firstly, different emotion categories exhibit varying performance across different values of K. For example, the “Surprise” category generally achieves higher precision, recall, and F1-score compared to other categories across all values of K. Secondly, for each emotion category, there seems to be an optimal value of K that maximizes the performance metrics. This optimal value may vary depending on the specific characteristics of the emotion category and the dataset. Also, adjusting the value of K can lead to trade-offs between precision and recall. For instance, increasing K may lead to higher recall but lower precision, and vice versa.

Table 5 Performance comparison of mutual information feature extraction method at various values of K.

K	50	80	100	120	150		
Metric	P	R	F1	P	R	F1	P	R	F1	P	R	F1	P	R	F1	S	
Angry	0.75	0.72	0.73	0.80	0.77	0.79	0.82	0.77	0.79	0.79	0.79	0.79	0.75	0.73	0.74	808	
Calm	0.39	0.27	0.32	0.44	0.32	0.37	0.42	0.53	0.47	0.43	0.82	0.57	0.66	0.63	0.65	93	
Disgust	0.47	0.45	0.46	0.51	0.50	0.50	0.51	0.53	0.52	0.59	0.54	0.56	0.52	0.51	0.51	843	
Fear	0.63	0.42	0.50	0.71	0.49	0.58	0.70	0.50	0.59	0.66	0.54	0.60	0.61	0.53	0.57	851	
Happy	0.62	0.50	0.55	0.69	0.54	0.61	0.66	0.57	0.61	0.67	0.62	0.64	0.57	0.56	0.57	865	
Neutral	0.53	0.54	0.54	0.58	0.64	0.61	0.63	0.62	0.63	0.64	0.65	0.64	0.55	0.59	0.57	709	
Sad	0.45	0.76	0.56	0.49	0.79	0.60	0.50	0.74	0.60	0.55	0.69	0.61	0.56	0.63	0.59	809	
Surprise	0.75	0.62	0.68	0.81	0.67	0.73	0.79	0.70	0.74	0.82	0.78	0.80	0.80	0.89	0.84	279	

Once again, emotion categories like “Calm” and “Sad” exhibit lower performance compared to others, which might be attributed to class imbalance issues or inherent difficulty in distinguishing these emotions from speech features alone.

In this approach, the best performance was obtained at a K value of 150 on the “Surprise” emotion, with a precision score of 0.80, a recall of 0.89, and an F1-score of 0.84. The poorest performance was observed at a K value of 50 on the ‘Calm’ emotion with a precision score of 0.39, a recall of 0.27, and an F1-score of 0.32.

Wrapper method results

Figure 7 and Table 6 represent the confusion matrices and table of performance of the recursive feature elimination wrapper-based approach implemented in this study. As observed from these results, like the MI approach, the different emotion categories exhibit varying performance across different values of K. For instance, “Surprise” tends to have higher precision, recall, and F1-score compared to other categories across most values of K. There also seems to be an optimal value of K for each emotion category that maximizes the performance metrics.

Figure 7 Confusion matrix for recursive feature elimination-based feature extraction for K = 50,80, 100, 120, and 150, respectively.

(A) K = 50 features. (B) K = 80 features. (C) K = 100 features. (D) K = 120 features. (E) K = 150 features.

Table 6 Performance comparison of recursive feature elimination feature extraction method at various values of K.

K	50	80	100	120	150		
Metric	P	R	F1	P	R	F1	P	R	F1	P	R	F1	P	R	F1	S	
Angry	0.72	0.70	0.71	0.72	0.74	0.73	0.76	0.76	0.76	0.77	0.76	0.76	0.73	0.74	0.73	808	
Calm	0.43	0.88	0.58	0.48	0.84	0.61	0.45	0.86	0.59	0.45	0.90	0.60	0.63	0.77	0.70	93	
Disgust	0.46	0.40	0.43	0.47	0.41	0.44	0.47	0.42	0.45	0.52	0.48	0.50	0.52	0.49	0.51	843	
Fear	0.61	0.42	0.50	0.58	0.44	0.50	0.62	0.46	0.53	0.66	0.50	0.57	0.60	0.50	0.55	851	
Happy	0.54	0.47	0.50	0.55	0.49	0.52	0.60	0.51	0.55	0.60	0.55	0.58	0.57	0.56	0.57	865	
Neutral	0.48	0.56	0.52	0.50	0.55	0.58	0.51	0.56	0.54	0.54	0.57	0.56	0.55	0.58	0.56	709	
Sad	0.49	0.70	0.54	0.51	0.67	0.58	0.50	0.69	0.58	0.53	0.70	0.60	0.55	0.65	0.60	809	
Surprise	0.68	0.64	0.66	0.78	0.72	0.75	0.76	0.72	0.74	0.83	0.77	0.80	0.87	0.86	0.87	279	

In this approach, the best performance is observed in the emotion category “Surprise” across all values of K, except 50. At K = 150, “Surprise” achieves the highest precision, recall, and F1-score among all emotions, with precision of 0.87, recall of 0.86, and an F1-score of 0.87. The poorest performance is seen in the emotion category “Disgust” across all values of K. At K = 50, “Disgust” achieves the lowest precision, recall, and F1-score among all emotions, with precision of 0.46, recall of 0.40, and an F1-score of 0.43. This implies that in the RFE feature extraction method, “Surprise” tends to be better distinguished and predicted compared to “Disgust” which faces challenges in accurate classification.

Statistical comparison

When conducting a statistical comparison of feature selection methods, it is essential to consider multiple evaluation metrics to understand the impact of each approach comprehensively. Table 7 compares various feature selection methods based on precision, recall, F1-score, accuracy, and the number of features selected. Figure 8 reveals that the best performance was attained at K = 120, where accuracy, precision, recall, and F1-score all peaked. Increasing the number of features beyond this point did not lead to further gains, indicating performance saturation. The accompanying color bar encodes the intensity of metric values: values closer to 0 are visualized in purple, while those approaching 1 are represented in yellow. This provides a quick visual reference for interpreting relative model performance across feature configurations.

Table 7 Result comparison of all feature selection approaches and baselines.

Feature selection approach	Precision	Recall	F1-score	Accuracy	No. of features	
All feature extraction methods	0.61	0.61	0.61	0.6142	170	
Five feature extraction methods	0.63	0.63	0.63	0.6294	163	
Six feature extraction methods	0.62	0.62	0.62	0.6225	157	
CB (0.2)	0.39	0.39	0.39	0.3899	7	
CB (0.1)	0.60	0.60	0.59	0.5967	133	
CB (0.05)	0.60	0.60	0.60	0.6037	150	
CB (0.02)	0.61	0.60	0.60	0.6045	158	
MI (50)	0.58	0.56	0.56	0.5615	50	
MI (80)	0.64	0.62	0.62	0.6159	80	
MI (100)	0.64	0.62	0.62	0.6232	100	
MI (120)	0.65	0.65	0.65	0.6471	120	
MI (150)	0.61	0.61	0.61	0.6058	150	
RFE (K = 50)	0.56	0.55	0.54	0.5478	50	
RFE (K = 80)	0.57	0.56	0.56	0.5627	80	
RFE (K = 100)	0.59	0.58	0.58	0.5779	100	
RFE (K = 120)	0.61	0.61	0.60	0.6053	120	
RFE (K = 150)	0.60	0.60	0.60	0.6036	150	

Figure 8 Performance of the CNN model across various values of K, using the MI feature selection method.

Metrics plotted include accuracy, precision, recall, and F1-score.

Using all 170 features as a baseline, the model achieved an average score of 0.61 across all metrics. With all 170 features included, this approach ensures that no potentially important feature is omitted. However, including all features can lead to the inclusion of noise and irrelevant features, which may not contribute to the model’s predictive power and could potentially degrade performance. This method serves as a reference point against which the effectiveness of feature selection methods can be measured.

When employing five and six-feature extraction methods, the model shows a slight improvement in performance metrics compared to using all features. The precision, recall, F1-score, and accuracy are slightly higher at around 0.63 for the five-feature method and 0.62 for the six-feature extraction method. The number of features selected is reduced to 163 and 157, respectively. These results suggest that reducing the number of features by removing irrelevant or redundant ones can enhance model performance. This improvement is likely due to a more focused model that can generalize better by training on the most informative features.

Correlation-based methods with different thresholds exhibit varied performance. The CB (0.2) method, with only seven features selected, shows poor performance across all metrics (0.39). This suggests that an extremely stringent threshold can significantly reduce the feature set, potentially omitting important features necessary for accurate predictions. On the other hand, CB methods with thresholds of 0.1, 0.05, and 0.02 maintain performance metrics comparable to the baseline (around 0.60–0.61) while reducing the feature set size to 133, 150, and 158, respectively. These results indicate that less stringent thresholds can help maintain model performance while improving interpretability and reducing complexity by excluding less relevant features.

MI methods generally show an improvement in performance as more features are included, with metrics peaking when 120 features are selected. The model achieves a precision, recall, F1-score of 0.65, and accuracy of 0.6471 with 120 features. This indicates that MI is effective in identifying the most informative features, and the optimal balance between feature reduction and model performance is achieved with 120 features. The performance of MI methods demonstrates that increasing the number of selected features beyond this point does not significantly enhance model performance, as seen with MI (150), which shows a slight decline in metrics.

RFE methods exhibit gradual improvement in performance as more features are included, peaking with 120 features. The best performance achieved is with precision, recall, F1-score at 0.61, and accuracy at 0.6053. This method’s iterative process of eliminating the least important features effectively identifies the most relevant ones. However, performance stabilizes around 120 features, similar to MI methods, suggesting a limit to the benefits of adding more features. This stabilization indicates that RFE is effective but does not benefit from excessive feature inclusion beyond a certain point.

As shown in Fig. 8, the overall best-performing method is MI with 120 features, achieving the highest precision, recall, F1-score, and accuracy. This method provides a balance between feature reduction and performance, making it the most effective in this comparison. Correlation-based methods with moderate thresholds (0.02 to 0.1) offer efficient feature selection with minimal performance loss, making them good options when model simplicity and interpretability are essential. RFE and MI methods are effective in identifying important features, with RFE showing steady improvements as more features are added.

In conclusion, choosing an appropriate feature selection method depends on the specific trade-offs between model performance and complexity. MI (120 features) stands out as the optimal method in this comparison, balancing feature reduction and high performance. CB methods with moderate thresholds also provide a good balance, improving model simplicity without significant performance loss. Iterative methods like RFE and MI are effective for identifying the most relevant features, demonstrating the importance of thoughtful feature selection in model development.

Conclusions

This article comprehensively evaluates various feature selection approaches for speech emotion recognition, comparing their performance across multiple metrics, including precision, recall, F1-score, and accuracy. By analyzing a range of methods, from using all available features to employing specific feature extraction techniques such as correlation-based methods, mutual information (MI), and recursive feature elimination (RFE), valuable insights into the effectiveness and efficiency of each approach were derived.

Our findings indicate that while using all features provides a solid baseline, it often includes noise and irrelevant data, potentially diminishing model performance. In contrast, targeted feature selection methods can significantly enhance model efficiency and accuracy. Among the evaluated methods, mutual information with 120 features emerges as the most effective, achieving the highest metrics across the board. This approach strikes an optimal balance between reducing the feature set and maintaining high model performance, making it a robust choice for feature selection.

Despite these positive results, there are several limitations that should be acknowledged. First, the dataset splitting was conducted using a random shuffling strategy at the audio sample level. As a result, samples from the same speaker may appear in both the training and testing sets. This may introduce speaker-dependent bias, potentially inflating the reported performance metrics. Furthermore, the use of data augmentation (e.g., pitch shifting and noise addition) alongside subject-dependent splits may have further amplified this bias, as augmented samples from the same speaker can resemble test data, leading to an overestimation of model performance. Future work will address these issues by adopting a subject-independent splitting strategy and carefully assessing the impact of data augmentation. Second, the datasets used are limited to English-language recordings, which may not generalize well across different languages or cultural contexts. Additionally, there is limited demographic diversity in terms of speaker age and gender, which may affect the robustness of the emotion recognition system in real-world scenarios. Emotions such as “Calm” and “Sad” also showed consistently lower performance, which could be attributed to class imbalance or the inherent difficulty in distinguishing these emotions based solely on speech features.

Future work will explore hybrid feature selection methods that combine filter- and wrapper-based strategies with representation learning. Additionally, extending the evaluation to multilingual datasets and incorporating more age-diverse and culturally varied speech corpora will allow for the development of more robust and inclusive emotion recognition models. Expanding the framework to include real-time and cross-lingual settings is also of interest, especially in applications such as virtual assistants and mental health monitoring.

Supplemental Information

Supplemental Information 1 Emotion dataset.

Supplemental Information 2 Emotion code.

Additional Information and Declarations

Competing Interests

The authors declare that they have no competing interests.

Author Contributions

Alaa Altheneyan conceived and designed the experiments, performed the experiments, analyzed the data, performed the computation work, authored or reviewed drafts of the article, and approved the final draft.

Aseel Alhadlaq conceived and designed the experiments, analyzed the data, prepared figures and/or tables, authored or reviewed drafts of the article, and approved the final draft.

Data Availability

The following information was supplied regarding data availability:

The CREMA-D (Crowd-sourced Emotional Multimodal Actors Dataset) data is available at GitHub: https://github.com/CheyneyComputerScience/CREMA-D.

The RAVDESS (Ryerson Audio-Visual Database of Emotional Speech and Song) data is available at Zenodo: Livingstone, S. R., & Russo, F. A. (2018). The Ryerson Audio-Visual Database of Emotional Speech and Song (RAVDESS) [Data set]. In PLoS ONE (1.0.0, Vol. 13, Number 5, p. e0196391). Zenodo. https://doi.org/10.5281/zenodo.1188976.

The TESS (Toronto Emotional Speech Set) data is available at: https://utoronto.scholaris.ca/collections/036db644-9790-4ed0-90cc-be1dfb8a4b66, https://hdl.handle.net/1807/24487.

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
