# Peer review of "Feature selection for emotion recognition in speech: a comparative study of filter and wrapper methods"

_PeerJ Computer Science, doi:10.7717/peerj-cs.3180_

## Round 0.1 · original submission · Major Revisions

Thank you for sending your valuable work to PeerJ in Computer Science.

While the manuscript addresses a hot topic, proposing a comparison of different methodologies for effective feature selection for speech emotion recognition, some important details are missing and require further description.

Considering the Reviewers' comments as well as the points reported in what follows, a major revision is suggested.

Please, provide a point-by-point response to all the Reviewers’ comments and particularly:

1. Clarify the differences between the first and second research objectives, e.g., by rephrasing the two points.

2. Consider expanding and/or highlighting the knowledge gap present in the literature.

3. Provide self-explanatory captions to all your figures.

4. Consider providing a map to follow the proposed methodology, that should be described in depth. Follow attentively Reviewer 1’s suggestions and answer the reported questions.

5. Provide a clear datum on the data dimensionality, specifying how much the data is augmented and if this operation is performed on the whole dataset or only on the training set to avoid possible biases.

6. Clarify how the statistical comparison is made, and address Reviewer 1’s concerns on table 5.

7. Add a discussion on the limitations of your approach as well as its future development.

·

Basic reporting

The authors in the article compare the using of various methods of features selection to improve the performance of a Speech Emotion Recognition Model. In the paper three different features selection methods were considered including Filter methods and Wrapper methods. In general, the manuscript is well written and easy to read. There are only few parts in the text that are ambiguous and should be better detailed. In particular, the definition of “all available features extraction methods” or “five or six features extraction methods” in the abstract should be better explained, as it is not clear which features are extracted by each of these methods. As a minor comment, there are some typos in the quotation marks (sometimes written as ”..” instead of “..”), as well as in the features names (e.g. in line 428 the SCF features extraction method is defined as SC).
The context of analysis and the performance achieved by the state of arts works are well described and detailed in section 1. The only exception is for the analysis of Likitha et al., 2017 (line 112) article where no value of accuracy is reported. For the sake of completeness, I suggest that the authors better describe also this work to emphasise the importance of considering also MFCC features in the definition of a Speech Emotion Recognition model.
Regarding the structure of the article, it is not clear to me whether the last part of section 2.1 ("Data collection and preprocessing") is included in the description of the RAVDESS dataset, or whether it is a general visual analysis of the speech audio signals in different emotional states.
Some figures lack details that make them difficult to understand and interpret. For example, it is not clear to me which dataset the histogram depicted in Figure 2 refers to, as well as the term “statistics” in the caption is a bit ambiguous and needs more explanation (e.g. what is the statistic reported in the graph?). Similarly, the meaning of the plot in Figure 8 is unclear. Which features selection method was used to achieve the performance reported in the graph? What is the meaning of the colorbar on the right-hand side of the picture?
Finally, as a minor comment, there is a black line at the top of some of the confusion matrices displayed in Figure 5 and 6 (e.g. Figure 5a, 5b, 5e, 6a, 6b, and 6e).

Experimental design

The research questions are well defined, although they sometimes seem to overlap. For example, the first and second research objectives seem to refer to the same goal, i.e. evaluating the performance of the SER classifier by considering different features selection strategies.
In addition, the description of the model needs more details and explanation too. Here are some observations:
1) In Section 2 (Methodology) it is not clear how the three datasets were combined to define the single dataset involved in the analysis. In particular, there is no information on how the authors took into account aspects such as different audio lengths, different sampling rates or even different emotions expressed by the speaker. Furthermore, a combined dataset is cited in the description of CREMA-D dataset, but it is unclear which are the datasets combined in this stage.
2) Likewise, it might be useful to report the number of audios recorded for each emotion in the three datasets considered. This might help to better clarify the problem of class imbalance, repeatedly mentioned in the Discussion and Conclusions section.
3) I suggest that the authors better describe how the data augmentation phase was applied in their analysis. In particular, it is not stated in the text to which data it was applied and how many new instances were created for each class. For the sake of completeness, I also recommend that the authors add information on the cardinality of the classes before and after the data augmentation phase.
4) In the Section 2.5.2 (Implementation Details) is reported that “the dataset was split by a proportion of 70:15:15 for the train, test and validation sets, respectively” but it is not reported how this segmentation was carried out. In particular, there is no information if the instances were randomly selected or considering a subject independent strategy.
5) In Section 3.2.2, different values of a “K” factor have been considered in the mutual information approach. What is this “K” factor? Is the number of features that you want to keep? Please, better detail this point in the text.
6) Finally, the thresholds reported in Table 5 for the Correlation Based features selection approach are not the same as those reported in Table 2 for the same algorithm. In particular, the performance achieved considering the threshold = 0.2 are not reported in Table 2 .

Validity of the findings

In general, the analysis performed is interesting, although it lacks some details that would allow for a proper reproduction.
Also, the conclusions are minimalist and could be improved by adding discussion of future studies or limitations of the current analysis. In particular, issues related to speech emotion recognition task, such as the use datasets with different languages or ages, could be reported in the conclusion.
Furthermore, I would also suggest the authors point out the limitations and bias of using data collected from the same subject to both train and test the model.

Reviewer 2 ·

Basic reporting

1) Recent Related works are not completely reviewed. More recent related works should be introduced to draw a big picture for the readers.
2) After the literature review, highlight in 9-15 lines what overall technical gaps are observed in existing works that led to the design of the proposed methodology.

Experimental design

The reference for performance metrics (Accuracy (ACC), Precision (P), Recall (R)) is missing in Section 2.5.3.

Validity of the findings

1) The authors should explain the drawback of the proposed work in the conclusion.
2) It would be better if we could see the possible research areas for future works to be recommended by the authors at the end of the manuscript.

---

## Round 0.2 · Minor Revisions

I thank the Authors for the revised version of their manuscript and the clear point-by-point response to the Reviewers.

The Reviewers are generally satisfied by the applied modifications and require just a minimum intervention on the current version of the manuscript.

Please, follow Reviewer 1 suggestions and particularly:

1- Clarify the doubts presented in the basic reporting section, especially in points 3 to 5.
2- Expand your discussion on the features selected by your methodology to better highlight the contribution on model interpretability stated in the introduction.

·

Basic reporting

The article is well described and detailed. There are only a few minor problems in the text that should be resolved:
1) [Abstract] Some features are listed in the abstract as acronyms (e.g., MFCC o ZCR). I suggest the authors include the full name of the features in the abstract as well.
2) On line 47 “Filter” is capitalized while “wrapper” is not. Similarly, there is no unique format for quotation marks in the text (see 'Surprise' at line 505 and “Surprise” at line 503). In general, I suggest that authors check the text to maintain a consistent writing style.
3) [Section 3.2.1] I do not understand the meaning of “blue” before each emotion (e.g., the blue “Surprise”): is this a typo or is it related to some particular figure/meaning?
4) [Section 2.3] I think it may be a typo in Formula 2: if S(n) is the log power at each of the mel frequencies, is the “log” in the formula correct?
5) [Section 2.2.4] I think that the last part of Section 2.2.4 (starting from “The original combined dataset…”) should be separated from the first part, as it is generic about cardinality after data augmentation and does not deal with the data incrementing technique Shifting.

Experimental design

no comment

Validity of the findings

In general, the analysis is well explained in the text. However, some parts could be better described and detailed:
1) I commend the authors for the good explanation in the conclusion about the possible limits and bias of the analysis. To this end, I suggest that the authors add in this part also a reference to the possible data augmentation bias. In particular, the use of a data augmentation strategy along with the use of a subject-dependent validation strategy may lead to increased performance because of the similarity between the data used to train and test the model.
2) Since the article deals with feature selection methods in the case of SER, I also suggest the authors to briefly comment what kind of features are generally retained by the features selection algorithm and which, on the other hand, are usually removed. This could also provide information on the importance of features in the context of emotion recognition from speech.

Reviewer 2 ·

Basic reporting

The paper is well documented and improved as per the suggestions. Thus, the revised version of the paper is well suited for publication.

Experimental design

Good

Validity of the findings

Good

Additional comments

Nil

---

## Round 0.3 · accepted · Accept

I thank the Authors for providing a clear point-by-point response to the Reviewers.

Please, notice that Reviewer 2 has not been involved in this round of revision, considering the Reviewer’s satisfaction with the previous manuscript version.

Reviewer 1 has positively accessed your final manuscript, and I have read through it to ensure that all the required modifications were applied.

The manuscript is now clear, well-written, and refined in all its parts. Therefore, it is ready for publication.

The PeerJ staff will surely help you in the final stages of publication. I suggest as a final remark enlarging Figure 5 to 7 for better readability.

Thank you for sending your manuscript to PeerJ in Computer Science and for your good work.

·

Basic reporting

The article is well-written and detailed, and it is ready for publication.

Experimental design

no comment

Validity of the findings

no comment